# Building More Resilient Culture Collections: A Call for Increased Deposits of Plant-Associated Bacteria

**DOI:** 10.3390/microorganisms10040741

**Published:** 2022-03-30

**Authors:** Kirk Broders, Andrew Aspin, Jordan Bailey, Toni Chapman, Perrine Portier, Bevan S. Weir

**Affiliations:** 1USDA, Agricultural Research Service, National Center for Agricultural Utilization Research, Mycotoxin Prevention and Applied Microbiology Research Unit, 1815 N. University, Peoria, IL 61604, USA; 2Fera Science Ltd., York Biotech Campus, Sand Hutton, York YO41 1LZ, UK; andrew.aspin@fera.co.uk; 3Biosecurity and Food Safety, NSW Department of Primary Industries, Plant Pathology & Mycology Herbarium, Orange Agricultural Institute, 1447 Forest Road, Orange, NSW 2800, Australia; jordan.bailey@dpi.nsw.gov.au; 4Biosecurity and Food Safety, NSW Department of Primary Industries, Elizabeth Macarthur Agricultural Institute (EMAI), Menangle, NSW 2567, Australia; toni.chapman@dpi.nsw.gov.au; 5Institut Agro, INRAE, Univ Angers, IRHS, SFR QUASAV, CIRM-CFBP, F-49000 Angers, France; perrine.portier@inrae.fr; 6Manaaki Whenua—Landcare Research, 231 Morrin Road, St. Johns, Auckland 1072, New Zealand; weirb@landcareresearch.co.nz

**Keywords:** accession, biobank, living collections, emerging pathogens, Biological Resource Centers, plant pathogenic bacteria

## Abstract

Biological collections preserve our past, while helping protect our future and increase future knowledge. Plant bacterial culture collections are our security for domestic and global biosecurity. This feature article will provide an introduction to the global position of plant bacterial collections. The role of collections in monitoring plant pathogenic bacteria will be explored through the presentation of five cases studies. These case studies demonstrate why culture collections were imperative for the outcome in each situation. We discuss what we believe should be the best practices to improve microbial preservation and accessioning rates, and why plant bacterial culture collections must increase deposits to be prepared for future emerging pathogens. This is not only the case for global culture collections, but on a much bigger scale, our future scientific successes, our biosecurity decisions and responses, and our knowledge are contingent upon preserving our valuable bacterial strains. It is hoped that once you read this article, you will see the need to deposit your strains in registered public collections and make a concerted effort to build better bacterial culture collections with us.

## 1. Introduction

If you are reading this, it is safe to assume that you know what a bacterial culture is. You have streaked plates, you have shaken broth, and you may even have hundreds of cultures tucked away on slants or in freezers. However, have you submitted these isolates to a registered culture collection?

Bacterial pathogens are a major threat to food and fiber production throughout the world. This can be observed in the destruction and impacts caused by *Xylella fastidiosa* to the olive crops in Puglia, Italy, and to the vineyards in the United States, or the effects of *Huanglongbing* (*Candidatus* Liberibacter spp.) on global citrus production. Global spread, the ability to wreak havoc in a short time, and our inability to respond quickly enough have become hallmarks of emerging bacterial plant pathogens around the world. The threat to food security is very real. Biosecurity networks and culture collections play a major role in combating these diseases. The need for collections is constantly increasing as we are faced with the movement of bacterial diseases within and between countries, the development of new strains, and the expansion of host ranges.

Discussion on the decline in plant collecting has been circulating for decades [1,2,3] and there have been many initiatives to improve herbarium resources, digitize specimens, and undertake targeted collecting trips. However, what of the microscopic, the often overlooked? Who will think of the bacteria? What is the rate at which novel bacterial strains are being deposited in culture collections around the world? More specifically, are plant pathogenic bacteria consistently being deposited in recognized and appropriate collections?

This article was born out of the alarming fact that there is a slow erosion of microbial collections the world over. We are seeing a decline in collecting and accessioning, in funding, staffing, and institutional support, and in teaching and career opportunities. All this is occurring when we have barely scratched the surface of the diversity, distribution, host range, ecology, and epidemiology of a number of emerging and re-emerging bacterial pathogens. We are also racing against time, as ecosystems are rapidly shifting due to human development and movement, the global distribution of plant material, and the impacts of climate change.

We are already facing the consequences of collection gaps. Projects seeking to perform phylogenetic and population studies using culture collections already have problems with gaps in data and the availability of cultures covering the true range of habitat, host, and location diversity of organisms. The development of rigorous diagnostic protocols is hampered by a lack of representation of native species. Collections are being accessed more than ever before, and efforts are being made to digitize records and make these data available online, but we are not seeing the same response in the deposition of cultures.

The World Data Center for Microorganisms (WDCM; http://refs.wdcm.org/, accessed on 27 March 2022) boasts 822 registered collections worldwide, with over 33 million cultures maintained by almost 7000 staff. This includes all culture holdings from plants, animals, and humans, and all microorganisms, but the WDCM’s statistics can still shed light on global trends in culture accessioning. Based on publicly available data from the WDCM collated in 2014 compared to 2021 numbers, Australia and New Zealand have had a 16% increase in cultures (which equates to ~2500 cultures a year), the USA has had a 40% increase (14,500 cultures a year), Europe a 17% increase (19,000 a year), the UK a 3.5% increase (3000 a year), while Asia is seeing a comparative boom with a 52% increase, with almost 500,000 cultures added to the directory, and China and India are leading this growth. However, many countries have seen enormous increases in recorded numbers of accessions with the WDCM (60–400% increases), and while this may be due to increased accessioning, it is also likely to be due to the digitization of records. Either way, this demonstrates a concerted effort to document, maintain, and provide access to these collections across 78 countries.

Estimates of the number of species of microbes on Earth range from millions to trillions [4,5,6]. The Global Biodiversity Information Facility (GBIF; https://www.gbif.org/ (accessed on 27 March 2022)) has almost 2 billion records of specimens and observational data, of which there are 19.5 million bacterial records (only 0.1%). IDigBio, which hosts only collection data with a physical specimen, holds 132 million records, of which 133,000 are bacteria (0.1%). Clearly, microorganisms are drastically under-represented in collections, given these species estimates. This is not surprising, because we cannot actually see them, and as humans we tend to be interested in, and support research on, the charismatic and the cuddly. The 2017 study on taxonomic bias in biodiversity data, which analyzed the GBIF data set, concluded that societal preference influences taxonomic research [7]. Think of all the attention and money that pandas and other cute mammals receive.

Collections are invaluable tools for microbiologists. Because they host many strains isolated at different times on different hosts or environments, and in different countries and continents, they summarize the collective sampling, research, and diagnostic efforts performed over time by bacteriologists from all over the world on many different bacterial species. These strains are important for understanding the epidemiology of a given species, when and where it was first isolated, and the historical prevalence of the species all over the world. However, collections harbor many strains with old names that no longer reflect their actual taxonomic status, as most were deposited before the advent of modern molecular taxonomy. This can seriously hamper the usefulness of the preserved strains for research. Collections are involved in huge efforts to improve this situation, but the isolates that need to be identified through molecular typing often number in the thousands or tens of thousands of strains.

While it remains an uphill struggle to convince the general public—or even funding bodies—that bacteria deserve attention, there is something researchers can do to help, and it just costs postage. It is called vouchering; the act of depositing strains and their related metadata in a recognized culture collection. Many journals require vouchering in order to publish, so you will have encountered this practice at some point, but as Editor in Chief for Application in Plant Sciences laments, authors keep asking, ’Why do vouchers even matter?’ [8].

Specimens have a myriad applications [9], but in this paper we offer a perspective for plant pathologists and regulatory scientists working with phytopathogenic bacteria in particular. We plan to convince you that all your hard work isolating and purifying should be captured in long-term storage at registered and dedicated facilities, for both your own benefit and for scientists to come. Below is a series of case studies based on several important and potentially destructive plant pathogenic bacteria that demonstrate why the phytobacteriology community should play an active role in depositing important research isolates into national and/or international culture collections.

## 2. The Role of Collections in Monitoring Plant Pathogenic Bacteria

***The case of Pectobacterium***. Members of the *Pectobacterium* genus cause soft rot in many plant species (including potato). They are characterized by their ability to degrade plant cell walls through the secretion of a cocktail of enzymes. The taxonomic history of this genus is complex, with numerous changes in the past 10 years (new species descriptions and elevation of subspecies to species level). Today 18 species are formally described and two others are proposed, 12 of them having been described less than 10 years ago [10].

The French Collection for Plant-associated Bacteria (CIRM-CFBP) preserves resources that are strategically important for plant health. Among them are numerous *Pectobacterium* strains isolated between 1944 and 2017, from all continents and from a great diversity of habitats. However, many of these strains were not correctly assigned to any of the described species (mostly due to the evolution of taxonomy). In response to this situation, the CIRM-CFBP clarified and updated the taxonomic status of 265 *Pectobacterium* strains in order to gain insight into their frequency and isolation habitat. They used a three-gene (*dnaX, leuS, recA*) multi-locus sequence analysis (MLSA) scheme (Figure 1) and complete genome sequencing, allowing the assignment of strains to clades and to assess the relationships between clades and species delineation [11,12].

This work resulted in several important findings. First, the majority of strains of the CIRM-CFBP collection belonged to a species described only in 2019 (*P. versatile*), even if some isolates were collected as early as 1946 (Figure 1), at which time scientists were monitoring the *Pectobacterium* population in crops and isolating a great number of them. However, further investigations systematically focused on the most aggressive strains, and because *P. versatile* was not among the studied strains we can hypothesize that this species has strains that are less aggressive on crops. Second, the taxonomic revision permitted a better understanding of the diversity of the genus. The species that were formerly grouped under the name *P. carotovorum* are closely related or are probably undergoing a speciation process, with average nucleotide identity (ANI) values at the border of the species limit. This work also uncovered putative new species remaining to be described [12]. Finally, the correct reclassification of all the strains permitted the revision of information associated with the different species. For instance, *P. peruviense*, named after the country where two strains were isolated, is actually more widely distributed, as shown by the fact that strains were also isolated in France. This allowed us to have a better understanding of the host range of all the species, as most of the strains in CIRM-CFBP are associated with the information of the host of isolation. For instance, *P. betavasculorum*, named after sugar beet and thought to occur only on this plant species, was also isolated from sunflower, potato, and *Opuntia* [12].

We now have a much better understanding of the *Pectobacterium* genus, its diversity, and the host range and geographical repartition of the different species. This was all made possible by the scientists who regularly deposited strains in collections. The efforts made by the collections to curate their resources are beneficial to the whole scientific community and bring a better understanding of the dynamics of the taxa considered. We strongly encourage culture collections to initiate or continue their efforts to update the identity of their resources.

***The case of Pseudomonas syringae pv. actinidiae***. Kiwifruit (*Actinidia* spp.) is an iconic New Zealand edible fruit. Although originally native to China, they were extensively bred in New Zealand in the early 20th century, with the more recent development of a ‘gold’ cultivar ‘Hort16A’ in 2000 [13]. Worldwide, kiwifruit are important economically, with production reaching up to 4.3 million tonnes/year and an estimated value of $US 7.6 billion worldwide [14]. In New Zealand, kiwifruit exports earn approximately $NZ 2.5 billion ($US 1.7 billion) per year, and they are the most important horticultural export crop by value [15].

Bacterial canker of kiwifruit was first recorded in California in 1980 [16], then in Japan in 1984, where it was described as a novel *Pseudomonas syringae* pathovar: pv. *actinidiae* (Psa) [17]. By 1992, the disease was found in Italy [18], but it was not a significant problem there until 2008, when kiwifruit orchards suffered heavy economic losses [19]. Genetic analysis of the strain from the 2008 Italian outbreak revealed that it was different from the previous 1992 strain and from the Japanese pathotype strain [20].

In November 2010, extensive necrotic leaf spots and wilting and browning of flowers was noticed on gold-fleshed kiwifruit in New Zealand. An initial diagnostic test [21] confirmed the presence of Psa. The International Collection of Microorganisms from Plants (ICMP) culture collection of New Zealand had the original pathotype strain of Psa, ICMP 9617, from Japan, for comparison, and as a positive control in diagnostic tests. Having this pathotype immediately available ensured there were no delays in the subsequent biosecurity response. Comprehensive mitigation strategies were used to limit the outbreak, including phytosanitary cleaning of all equipment moving between orchards, and in some cases orchard destruction, which was also implemented in the outbreak in Italy and France.

The origin of the New Zealand outbreak was unknown, with some kiwifruit growers suggesting they had seen leaf spotting in the past. To address this question, all *Pseudomonas* cultures in the ICMP isolated from kiwifruit were screened with the diagnostic primers, cultures that dated back to the 1970s. None of these isolates tested positive for Psa, suggesting the pathogen was not historically present in New Zealand [22]. However, the deposits of *Pseudomonas* spp. isolates from kiwifruit into the national ICMP collection had been sporadic, and prior to the 2010 outbreak the most recent deposit was in 1997, a 13-year gap (Figure 2). The lack of regular deposits of bacterial pathogens from New Zealand’s most important horticultural export crop limited the extent of the historical investigations that could be done.

An in-depth study was conducted on the global populations of Psa, with the diverse cultures stored in the several global culture collections critical to enabling this research. This revealed four global populations [23] (Table 1). Multigene sequencing of Psa isolates revealed that there were two variants in New Zealand: the outbreak-virulent strain ‘Psa-V’ (=Psa3) and a less virulent strain ‘Psa-LV’ (=Psa4) [22]. With Psa-LV, the symptoms did not progress beyond leaf spotting. The Psa outbreak in New Zealand affected the country’s most important export crop and had a significant economic and social impact, with one estimate of the total loss of equity for the country as high as NZ$2 billion [24].

In Italy and France, the Psa outbreak spread further, leading to the implementation of severe measures in order to limit the disease expansion (i.e., orchard destructions). However, some orchards suffered only mild damage. The symptoms did start as typical Psa symptoms (leaf spots), and were detected as such, but they did not evolve to cause severe damage and the trees were not severely affected. An extensive study of the French isolates showed that the French orchards were in fact infected by two closely related but phylogenetically distinct pathogens. The strains causing only mild symptoms were then described as a new pathovar, *P. syringae* pv. *actinidifoliorum* (=Psa4) [25]. This prevented serious crop loss, permitting regulatory professionals, scientists, and growers to focus on the orchards affected by *P. syringae* pv. *actinidiae* (=Psa3), which was a relief for producers. This again demonstrates that biological resources are crucial to determine whether a pathogen is endemic and has adapted, or represents a novel incursion.

The presence of type and reference cultures at ICMP enabled a rapid diagnostic response and delimitation of pathogenic populations. The presence of historical cultures allowed some insight into the origins of the pathogen, but inconsistent collections over time limited this use. We recommend that reference strains of pathogens of a country’s high-value crops be kept in a local culture collection, and that regular deposits of isolations of bacterial pathogens be made.

***Dickeya in pineapple—the Australian story***. *Dickeya*, like many phytobacteria, has had a very complicated taxonomic life. Originally described as *Bacillus* [26], in 1917 this was changed to *Erwinia* [27] after Erwin Frink Smith, who played a major role in the discovery of phytobacteria. The name *Erwinia chrysanthemi* was assigned in 1953 by Burkholder [28], but it was reassigned as *Erwinia carotovora f* sp. *zeae* a year later [29]. In 1980 Dye presented it as *Erwinia chrysanthemi* pv. *zeae*. A major revision of the *Erwinia* genus in 1998 resulted in the soft-rotting *Erwinia* spp. being reassigned to the genus *Pectobacterium* [30,31]. Samson et al. [32] further reclassified *Pectobacterium chrysanthemi* into six species of a new genus, *Dickeya: Dickeya chrysanthemi*, *Dickeya dadantii*, *Dickeya dieffenbachiae*, *Dickeya dianthicola*, *Dickeya zeae*, and *Dickeya paradisiaca*. *D. dieffenbachiae* has since been reclassified as a subspecies of *D. dadantii* [33]. *D. zeae* strains are so diverse that isolates group into two clades, with an ANI of 94–95% similarity, indicating that they should be two separate species [34].

Now in the era of gene sequencing and whole-genome sequencing, we are learning more about these pathogens and their relatedness, which may have an impact on their taxonomy, with potential biosecurity implications. This makes bacterial collections even more important: even if the originally deposited taxonomy is outdated, the strains are still available for identification using new technologies when needed. This case study is one example where two extremely valuable culture collections were used to prevent a full-scale biosecurity response. It is the story of the *Dickeya zeae* detection in Australian pineapples.

The bacterial genus *Dickeya* is a group of phytopathogens responsible for causing soft rot in a diverse number of hosts. The initial disease symptom for this soft-rotting pathogen is discoloration of the leaf sheath. This discoloration appears as water soaking, which progresses through the stalk and leaf tissue, resulting in the final collapse of the plant. Soft rotting of pineapple leaves was observed in a Queensland plantation in 2016. The symptoms appeared as classical bacterial heart rot of pineapple, of which the causative pathogen is *Dickeya zeae*.

Isolation of the causative agent then required identification, for which the MLSA protocol was used [35]. *Dickeya zeae* had not previously been reported in Australia, so there were no reference isolates for comparison, only MLSA gene region sequences available online. The value of the Australian NSW DPI Plant Pathology and Mycology Herbarium (DAR) and Queensland Plant Pathology Herbarium (BRIP) collections was highlighted, as all deposited *Erwinia* and *Dickeya* isolates were recovered as part of this investigation. Within the DAR and BRIP Australian collections, there were 16 *Erwinia* dating back to 1959, and 86 *Dickeya* (a number of isolates were renamed based on this study) dating back to 1978. Hosts included banana, potato, celery, onion, ginger, chrysanthemum, *Dieffenbachia*, and later pineapple, plus samples collected from river systems.

These accessioned isolates were recovered and sequenced with the *recN* gene [35]. Alignment and analysis of these sequences revealed that the isolate (BRIP64263) was *Dickeya zeae*, but it did not group with previously identified international isolates [35]. MLSA of the *dnaA*, *dnaJ*, *dnaX*, *recN*, and *gyrB* were conducted on the pineapple isolate and the isolates it grouped with, which revealed that this was not the same isolate as those from pineapples in Hawaii. The comparison of the MLSA revealed that in fact the pineapple isolate (BRIP64263) was the same as an isolate (DAR73906) that had been causing soft rot in ginger in 1998 in the same geographical location. This discovery prevented a full-scale biosecurity response and resulted in substantial cost savings, which could not have occurred without the historical bacterial collections.

However, due to the time involved in the recovery of the isolates and molecular analysis and sequencing, plus the outdated isolate taxonomy from the historical collections, trace-back and trace-forward of plant material had begun. Additional plants displaying soft rot symptoms were detected in both the Northern Territory (NTPCD43521, 43531 & 43539) and Queensland (BRIP65663 & 65665). Analysis of these isolates revealed they were also *Dickeya zeae*, but again they were shown by MLSA to be different from the original pineapple *Dickeya zeae* (DAR64263). The isolates from the Northern Territory plantation were genetically similar to a banana isolate from Queensland (DAR64262). The pineapple plantation in the Northern Territory had previously been a banana plantation, and bananas were no longer able to be grown due to a banana freckle incursion.

Again, what the collection isolates were able to provide was information on their current endemic presence within Australia. This in turn prevented a biosecurity response, as the molecular comparisons revealed these isolates were closely related to several domestic *Dickeya* spp. deposited in the collection over the years. The additional value of the collection was demonstrated with the ability to use the historical isolates from ginger (DAR73906) and run these in parallel with the pineapple isolates (BRIP64263 & NTPDC43521 & 43539) in pathogenicity assays with both ginger and pineapple. Without the collection, this would not have been possible. For biosecurity purposes these bacterial collections are invaluable, because they provide a baseline or starting point for known pathogens that are present within a country and the hosts they are present in. From a researcher’s perspective, they are a goldmine to explore.

***Citrus canker—another Australian case study***: *Xanthomonas citri* subsp. *citri* (Xcc) is the pathogen responsible for the devastating disease of citrus known as citrus canker. Currently listed as exotic to Australia, there have been three incursions into the Northern Territory, in 1912, 1991 (subsequently identified at a second location in 1993), and 2018, and two incursions into Queensland, in 1984 and 2004 (www.business.qld.gov.au/industries/farms-fishing-forestry/agriculture/crop-growing/priority-pest-disease/citrus-canker, accessed on 27 March 2022). All incursions were successfully eradicated. A number of these incursion isolates have been stored in the DAR reference collection, and they include the Northern Territory isolate from 1991 (DAR84834), the Northern Territory isolates from 2018 (DAR84953–84956), and those from Queensland in 2004 (DAR84952 & DAR84847).

Prior to Australia implementing strict biosecurity rules for the importation of exotic plant pests, a global collection of *Xanthomonas citri* subsp. *citri* was imported, stored, and accessioned in the DAR collection. This collection consists of 32 global isolates, seven Australian incursion strains, and three international border intercepts. This collection ranged in dates from 1988 to 2018, but it wasn’t accessioned until 2020. The origins of these isolates include the USA, Oman, Saudi Arabia, Iran, Thailand, Torres Strait, East Timor, and South-East Asia. There are whole-genome data for the isolates in this collection, which were used to develop species-specific diagnostic assays for the detection of *Xanthomonas citri* subsp. *citri* in Australian conditions. This assay was incorporated into an updated National Diagnostic Protocol (NDP-9 Asiatic citrus canker *Xanthomonas citri* subsp. *citri* V1.2), which was submitted at the end of March 2018. The following weekend, in April, a lime tree was observed in a retail store as being likely to have been infected with citrus canker. The samples were processed with the existing NDP and the new diagnostic assay. A crucial question was raised regarding this new incursion isolate, because the origin of the detection was in a similar location to the 1991 strain. Was this new isolate the same as the 1991 incursion isolate, and had it remained undetected in the environment?

The Xcc collection contained the previous incursion isolates from the Northern Territory (1991) (DAR84834) and Emerald (2004) (DAR849528 & 849528), providing a baseline for comparison. The genome sequence of this isolate was compared to the genomes in our collection. The new incursion isolate was identified as the A* pathotype. All previous Australian incursion isolates that had been deposited into the DAR had been the Xcc A pathotype. This new isolate, plus the two previous incursion isolates, were then able to be used in pathogenicity assays to confirm the host range variation between this strain and the 1991 strain. This pathogenicity assay would not have been possible without the preserved historical DAR collection. This is also true for the genomic studies, which would not have been possible without the historical collection, and nor would the development and testing of the diagnostic assay.

***The case of bacterial leaf streak of maize—Xanthomonas vasicola pv. vasculorum***. Bacterial leaf streak of maize was officially reported in the United States in 2016, but it had probably been present for several years prior to the official announcement from the US Department of Agriculture’s Animal and Plant Health Inspection Service (USDA-APHIS). The pathogen had spread rapidly throughout much of the semi-arid production regions of Colorado, Kansas, and Nebraska, concerning growers and international trade partners alike. Given the uncertain origin of the pathogen, a rapid and accurate identification was needed.

There were multiple hurdles that needed to be overcome that slowed the official identification of the causal organism, and this had real-life trade implications. One of the first hurdles consisted of the taxonomic issues associated with this pathogen. The bacterium *Xanthomonas vasicola* pv. *vasculorum* had previously only been reported in South Africa. The isolate, NCPPB 206, used to describe the disease in South Africa was determined at the time to be *Xanthomonas campestris* pv. *vasculorum*, and deposited in the National Collection of Plant Pathogenic Bacteria (NCPPB) in the United Kingdom in 1948. At that time, the disease caused limited damage and was generally thought to be of limited economic importance [36].

The disease was not reported again until the mid-1980s, at which time the bacterium was referred to as *Xanthomonas campestris* pv. *zeae*, to distinguish these isolates from those infecting sugarcane, *X. campestris* pv. *vasculorum*, and those infecting sorghum, *X. campestris* pv. *holcicola* [37,38]. Although these strains were not deposited in a culture collection, they were maintained by Dr. Larry Claflin at Kansas State University. The species *X. vasicola* was later proposed for *X. campestris* pv. *zeae*, *X. campestris* pv. *vasculorum*, and *X. campestris* pv. *holcicola* based on DNA-DNA hybridization [39] and fatty acid profiling [40] data, with both maize- and sugarcane-infecting strains combined into *X. vasicola* pv. *vasculorum*, and sorghum strains into *X. vasicola* pv. *holcicola*. However, a type specimen for Xvv was never designated and the name was not legitimate.

As a result, when a *Xanthomonas* species causing bacterial leaf streak symptoms first appeared in diagnostic labs in the U.S. in the summer of 2014, it was unclear where this bacterial pathogen had come from. The initial report by Korus et al. [41] merely stated that the bacterium causing the leaf streak symptoms on corn was *X. vasicola*, but did not indicate the pathovar. *X. vasicola* pv. *holcicola* is native to North America and is known to infect sorghum, but it had not been reported on maize. This left two potential scenarios: either Xvv had been recently introduced to the U.S. from South Africa, or an isolate of Xvh from the native population had made a host jump to maize. As mentioned, there were no isolates of Xvv or Xvh present in culture collections in the U.S., and only one strain of Xvv from maize in the NCPPB that had been collected in 1948, and one strain of Xvh in the NCPPB from the U.S., collected in 1961. Fortunately, the research collection of Dr. Claflin was preserved by Dr. Jan Leach at Colorado State University, and contained maize isolates of Xvv (originally *X. campestris* pv. *zeae*) from South Africa collected in the 1980s, and multiple isolates of Xvh from sorghum in the U.S. It was these isolates, in combination with contemporary strains, that were used in comparative genomics and pathogenicity/host-range studies to determine that the pathogen causing bacterial leaf streak of maize in the U.S. was Xvv, the same pathogen causing the disease on maize in South Africa [42,43]. It was also the work of Lang et al. [42] that finally established *X. vasicola* pv. *vasculorum* as an official name, and deposited strains of Xvv from maize in the U.S. and South Africa in the NCPPB.

The question remained as to how a pathogen of limited importance from South Africa arrived in the U.S. and quickly spread throughout the semi-arid corn-growing region of the U.S., resulting in disease incidence levels above 90% in many fields. What had changed in the 30 years between the time when the last known isolates of Xvv from maize were collected in South Africa and the point of emergence of the disease in the U.S.? Unfortunately, there are no isolates of Xvv from 1988 to 2016. However, after the publication of the first report by Korus et al. [41] and the taxonomy of Xvv by Lang et al. [42], researchers in Argentina realized that the disease they had been observing on corn in their country may also have been bacterial leaf streak caused by Xvv. While reports of severe symptoms similar to bacterial leaf streak on maize had been reported in Argentina since 2010, isolates of the causal organism were not collected and preserved until 2015 [44].

Researchers from Argentina, South Africa, and the U.S. worked together to sequence the genomes of multiple isolates for Xvv from all three countries to help track the global spread of this pathogen. They found that isolates of Xvv had probably been introduced into the U.S. and Argentina on two separate occasions [45]. They also found that all isolates of Xvv from the U.S. and Argentina contained a ~5 Kb fragment of prophage-related DNA that is also shared by Xvh strains, and was probably acquired by Xvv from Xvh through horizontal gene transfer [45]. There was only a single South African Xvv isolate that had this prophage gene cluster, but unfortunately, although the genome sequence is publicly available in GenBank, the isolate was not deposited, and the strain no longer exists. The importance and origin of this prophage region to the increased virulence of Xvv isolates from the U.S. and Argentina is still unclear.

Depositing strains of pathogenic species that cause common diseases, even if thought to be of minor economic significance, will continue to be of great importance to understanding the next emerging pathogen, as global trade and movement of seed continue to expand. In addition, while genome sequence data can be safely stored on hard drives and backed up in cloud-based storage, there are limitations to the inferences that can be drawn from these data if the biological organism from which they were generated is not also ‘backed up’ in a culture collection.

## 3. Best Practices to Improve Microbial Preservation and Accessioning Rates

As the above case studies demonstrate, the world’s culture collections have played an important role in the detection of new and emerging plant diseases, but many gaps in these collections remain and limit their utility. There is a need to improve the temporal and geographical representation of phytopathogenic bacteria in culture collections. There has been a decrease in the accession rate of bacteria in general, and bacterial plant pathogens specifically, over the last 50 years (Figure 3). These gaps represent significant challenges when trying to determine whether a new disease is caused by an endemic or introduced pathogen.

We believe this is due, in part, to the research community being unaware of this important issue, and there needs to be a shift in how scientists think about their biological research material. We propose that it be treated much like the sequence and whole-genome data generated from many of these strains. Specimens should be deposited in registered culture collections in the same way sequence data is deposited in registered databases, such as GenBank http://www.ncbi.nlm.nih.gov/genbank/, accessed on 27 March 2022, DNA Data Bank of Japan (DDBJ) (http://www.ddbj.nig.ac.jp/index-e.html, accessed on 27 March 2022), and EMBL Nucleotide Sequence Database (EMBL-Bank) (http://www.ebi.ac.uk/embl/, accessed on 27 March 2022). Below we provide a list of best practices that researchers, scientific societies, scientific journals, funding agencies, and federal governments should consider for their laboratories, research projects, funding programs, and publishing guidelines.

***Develop a specimen management plan***. Early in the research-design phase of a project, scientists should already be developing a specimen-management plan. Research budgets should include funds for accessioning costs, as this should be considered an expense of the research project. Funding agencies should require a biological specimen management plan for grant proposals that outline how new biological material and associated metadata will be preserved, similarly to a data-management plan. Moreover, a specimen-management plan is in line with access- and benefit-sharing regulations (related to the Nagoya protocol) and will help research organizations to comply with international regulations.

***Make microbial specimens available to the scientific community***. Journals should require that strains used for a new disease report, associated with a publicly available or published whole-genome sequence or used in the development of diagnostic protocols, be deposited in at least one culture collection. This is already included in standards for the development of diagnostic tools (for instance, EPPO 2020 [46]). Additionally, journals can require this of authors, rather than it being merely a *suggestion*. Researchers are strongly advised to deposit strains in advance, so that they can use culture collection accession numbers in their publications, improving future consistency.

***Collect, record, and safeguard specimen metadata***. When depositing new strains, provide the collection with as much metadata as possible associated with the strains. This provides a centralized location for all metadata associated with a strain, increasing the biological value of the strain for future diagnostic and research purposes. Moreover, for regulation purposes, some information is crucial, such as the country of origin (in general, this is the country of isolation) or the funding organization of the isolation. Organizations must strive for higher standards regarding the accountability of specimens collected and/or used by their researchers. Specimen acquittal procedures should be included in project proposals, and there must be oversight to ensure they are followed.

***Secure currently held biological resources***. Researchers who will be retiring or leaving their institution should work with their institutions to coordinate with a culture collection 1–2 years in advance, to ensure much of their collection is preserved for future scientists, as most institutions will not retain this material. If a researcher has amassed a large research collection, they should consider accessioning strains into a culture collection immediately in order to reduce the burden of depositing several hundred or thousands of strains at once.

In general, we strongly encourage scientists to deposit their important resources in public culture collections, where they will be made available long term for the benefit of the whole scientific community. Because the quality of the resources is dependent on the quality of the associated data, depositors are also strongly encouraged to transmit the most accurate and complete data, even if these data do not appear to be important at the time of deposit. This may seem like additional work for research teams, but there are benefits and incentives for researchers to deposit their strains with a culture collection, which they may find are worth the effort (see Figure 4).

## 4. Conclusions

The significance of existing and yet-to-be-discovered bacterial plant pathogens is shown in this paper. Their importance will increase as the world undergoes inevitable climatic changes. As host plants become established in new territories, there will be new interactions with the pathogenic bacteria that are already present, and pathogens will be spread into new geographical ranges. The problem is that there is a need to accelerate the rate at which data are collected and used to answer the questions the future will bring, but the number of depositions of plant pathogenic bacteria in the relevant collections is actually decreasing (Figure 3). Further, as has been outlined, we do not start from the best of positions: bacteria are among the most diverse and abundant organisms on and in the earth, yet they are extremely poorly represented in the world’s collections. For epidemiological studies, isolates representing as complete a span of time and as comprehensive a range of geographical locations and environments as possible are needed. We need diagnosticians and researchers to not only continue depositing strains in public collections, but also to redress the balance and increase the rate of deposition. For the benefit of the scientific community, publicly available, long-term storage of these organisms in dedicated specialist facilities is required.

Our detailed case studies highlight the need for such facilities. Work carried out in our collections, utilizing strains deposited over many years and using new techniques, has: clarified once unanswerable questions related to the host and geographical range of a problematic genus; ensured a biosecurity response was proportionate and timely, and, using strains from other collections, filled knowledge gaps and determined variability within a species; provided baselines to make comparisons in outbreak situations, and highlighted the importance of accurate taxonomy and nomenclature; utilized strains with enough diversity housed in a collection to enable the development of a national diagnostic assay; used a historical strain, isolated from Africa and housed in Europe, that was integral in trying to resolve an outbreak in North America.

For each of these success stories, there are gaps in knowledge that, if filled by isolates that were probably considered unimportant at the time, or were thrown away as part of a private collection when the curator ceased to have an input, would have made the journey so much easier and/or quicker. Without the collection of pathogenic species on a periodic basis, making comparisons with emerging pathogens is greatly hampered. For every success story, there could be tens, or hundreds, or possibly even thousands of ongoing and future bacterial plant pathogen questions that, unless there is an increase in accessible, curated, well-preserved accessions, might not be answered because of the lack of data. It is optimistic to think that the situation can be resolved by capturing the genome sequence of these organisms, but, as our last case study highlights, there are traits that cannot be inferred from sequence data. The ultimate question is, ’How does a bacterial plant pathogen interact with its host(s) in differing conditions?’, and sequence data might never be able to tell us that.

We have outlined in our list of best practice recommendations the importance of journals and funding agencies requiring depositions of strains into public collections, the importance of bodies responsible for diagnostic assay development and researchers utilizing publicly accessible cultures from collections, and the importance of culture deposits being accompanied by extensive metadata. We also ask that enough time be allowed when transferring privately held collections to a public collection. The benefits are highlighted in Box 1—both for you and for the scientific community.

In the introduction to this review, we asked some questions. We hope that having read it, you will want to deposit your strains in registered public collections, and you will make a habit of utilizing collections that specialize in plant pathogenic bacteria–for the benefit of everyone. We close by thanking those who have taken the time to deposit their cultures, whether throughout history or only this week.

## 5. Where Should I Deposit?

In the world there are numerous public collections, generalist or specialized in a particular topic, in which to deposit interesting microbial strains. The majority of them are registered at the WFCC, the World Federation of Culture Collections (http://www.wfcc.info/ (accessed on 27 March 2022)). Many collections can also assist you with biosecurity rules and regulations. The global collections, such as the ARS Culture Collection, CBS, and ICMP, have permits that allow them to import plant pathogenic bacteria from nearly any country in the world, and they can also work with you to determine whether your country requires an export permit. International catalogues for biological resources are also available, e.g., GBIF, Global Biodiversity Information Facility (https://www.gbif.org/ (accessed on 27 March 2022)), or WDCM the World Data Center for Microorganisms (http://refs.wdcm.org/ (accessed on 27 March 2022)). In Europe ECCO, the European Culture Collections’ Organisation regroups the majority of European microbial collections (https://www.eccosite.org/ (accessed on 27 March 2022)). In Europe, the Microbial Resource Research Infrastructure MIRRI (https://www.mirri.org/ (accessed on 27 March 2022)) is being built, and a general catalogue, along with other services, is already available.

Table 2 below lists culture collections specialized in plant-pathogenic bacteria. Through these collections, the data on cultures cited in this article are publicly available.

## Figures and Tables

**Figure 1 microorganisms-10-00741-f001:**
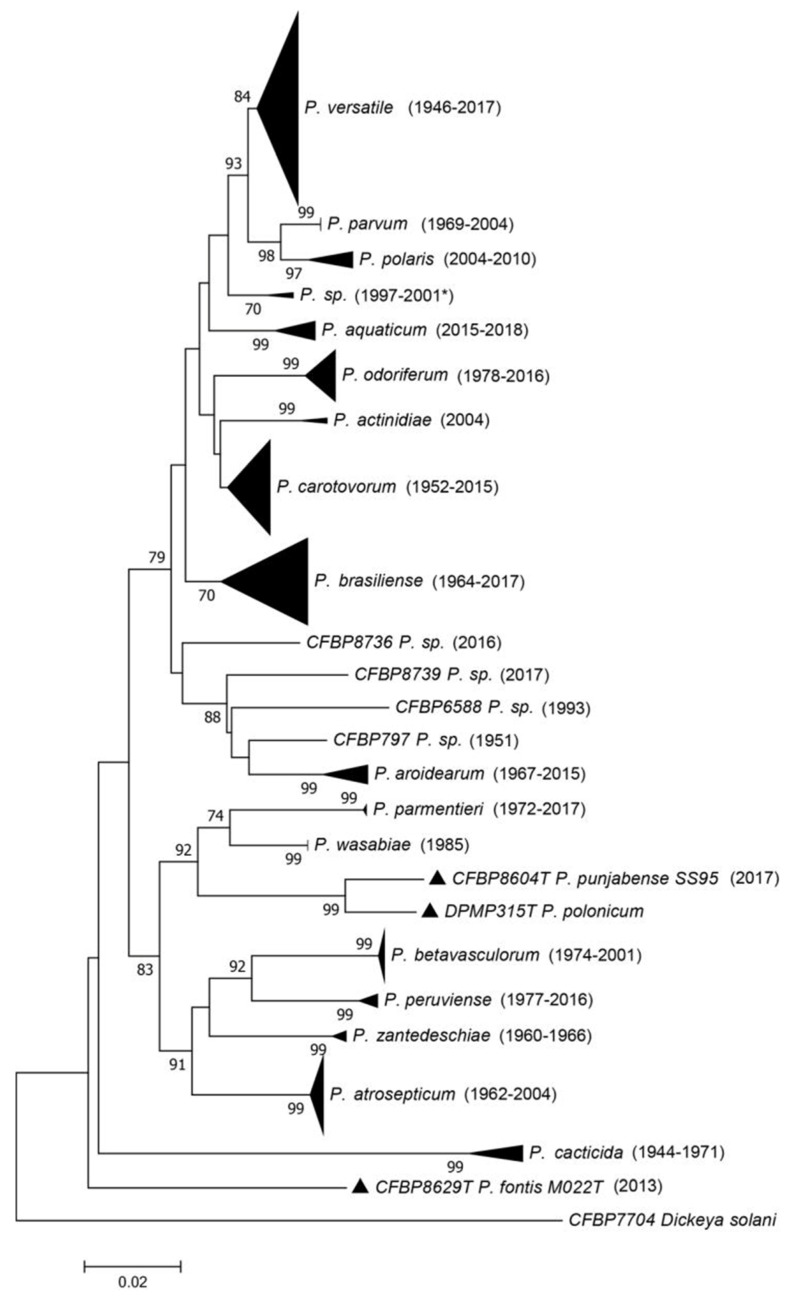
Phylogenetic tree reconstructed from concatenated partial sequences from *dnaX*, *leuS* and *recA* housekeeping genes. The phylogenetic tree was reconstructed with concatenated alignments of all genes with MEGA 7.0.26 using the neighbor-joining method with 1000 bootstrap replicates, and the evolutionary distances were computed using the Kimura two-parameter method. Bootstrap values are shown when over 70. The earliest and latest isolation years for each species are shown in parentheses. When there is only one strain in the clade, the year of isolation for that strain is indicated. For one clade, the latest year of isolation is unknown, and the latest year of deposit is indicated as follows: 2001*. Source: Portier et al. [12].

**Figure 2 microorganisms-10-00741-f002:**
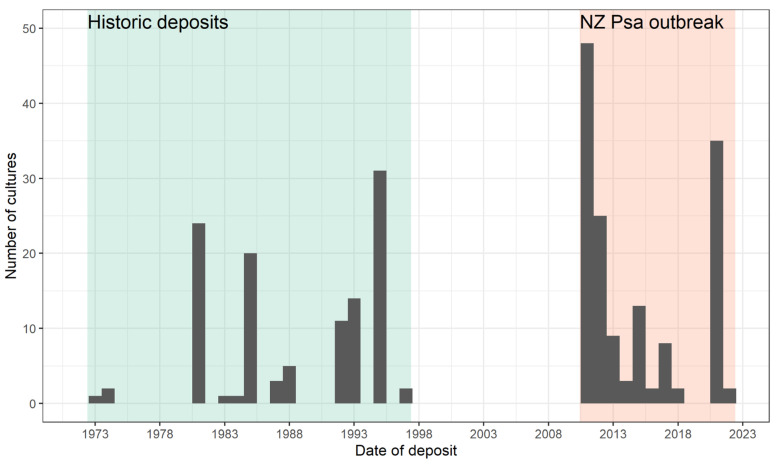
Deposits of *Pseudomonas* cultures isolated from New Zealand kiwifruit in the ICMP culture collection. A 13-year gap in deposits before the 2010 outbreak limited the use of the collection in determining the timeline of the origin of the incursion.

**Figure 3 microorganisms-10-00741-f003:**
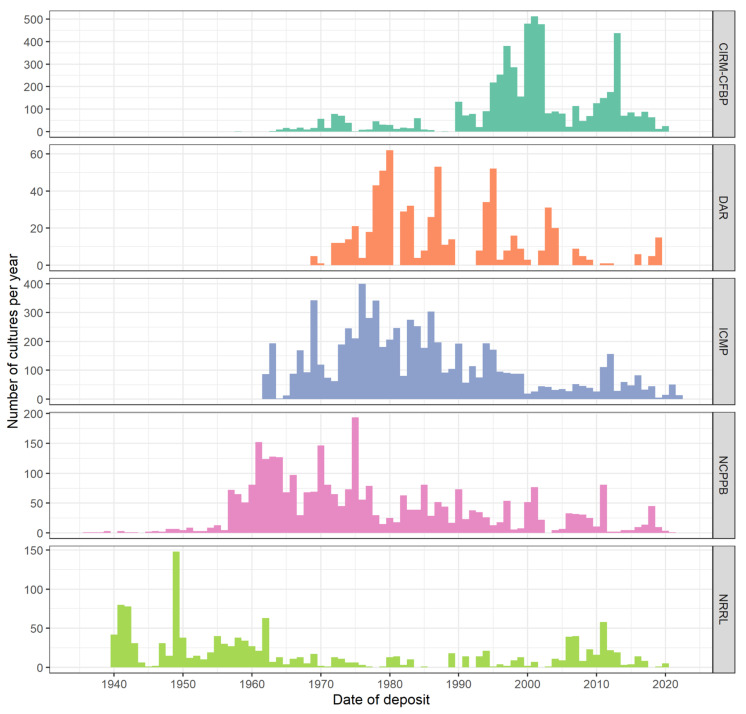
A histogram showing the dates of strain deposit in five global phytopathogen culture collections: CIRM-CFBP, DAR, ICMP, NCPPB, and NRRL. The collections have different establishment dates reflected in the first deposit of cultures, but all show a pattern of a reduction in deposits in recent decades from historical patterns.

**Figure 4 microorganisms-10-00741-f004:**
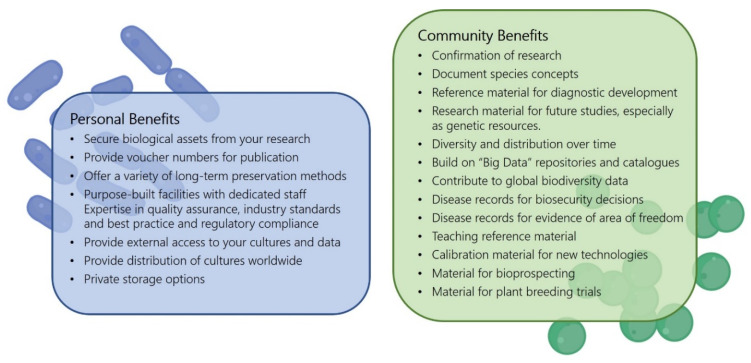
Benefits of depositing strains as based on what we learned from the case studies.

**Table 1 microorganisms-10-00741-t001:** The changing nomenclature of global populations of *P. syringae* pv. *actinidiae*.

Population [23]	Country and Year of First Report	NZ Name	Pathovar [25]
Psa1	Japan (1989)		*actinidiae*
Psa2	Korea (1994)		*actinidiae*
Psa3	Italy (2008)	Psa-V	*actinidiae*
Psa4	New Zealand (2011)	Psa-LV	*actinidifoliorum*

**Table 2 microorganisms-10-00741-t002:** International culture collection that specialize in accessioning and distributing plant-associated bacteria from countries around the world.

Collection	Country	Web Address (All Accessed on 27 March 2022)
CIRM-CFBP	France	https://cirm-cfbp.fr
DAR	Australia	https://www.dpi.nsw.gov.au/about-us/services/collections/collection-services
ICMP	New Zealand	https://www.landcareresearch.co.nz/icmp
NCPPB	United Kingdom	https://www.fera.co.uk/ncppb
NRRL	United States of America	https://nrrl.ncaur.usda.gov/

## Data Availability

The R code used to generate figures in this study is openly available in GitHub at: https://github.com/onco-p53/manuscripts/tree/main/culture-collections (accessed on 27 March 2022).

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
