# Peer review of "Building More Resilient Culture Collections: A Call for Increased Deposits of Plant-Associated Bacteria"

_microorganisms, 2022, doi:10.3390/microorganisms10040741_

Round 1

Reviewer 1 Report

Manuscript entitled “Building more resilient culture collections: A call for increased deposits of plant-associated bacteria” is an interesting perspective article, where authors describe the needless of a universal, well documented, and fully available collection of microbes (in this case focalized in plant pathogenic bacteria). Authors describe the advantages of the existence of these type of collections in order to determine de origin and evolution of plant disease produced by bacteria. They illustrate with several cases of these diseases and how the existence of a bacterial collection could give more information about the origin, spreading, host specificity of a plant pathogenic bacteria, and the probable future behaviour of this disease.

I am fully in agreement with the content of this manuscript; described microbes with a agricultural potential must be deposited in a/several universal collections and must be fully and free available for scientists. This is the way to study microbes isolated in the past and give the opportunity to apply on it the new scientific techniques in order to clarify the evolution of a concrete pathogen/beneficial microbe from the past to the present moment and this fact could help to determine the future of a plant disease and its biocontrol in the changing climate conditions that we are suffering.

I can describe my case; I would like to work with a devastating plant pathogen, in a laboratory with the necessary security conditions, in order to apply a new biocontrol strategy against this microbe, but at this moment a “lobby” working on this microbe exists and I have a lot of difficulties to acquire this strain. The existence of these universal a free collection/s can facilitate the work of a lot of scientists.

I think that the only thing that I can propose to the authors is the mentioning of the necessary requirement of the biosecurity rules for the bacterial acquisitions from these general and universal microbe collections.

Author Response

We thank the reviewer for their kind comments. Based on their suggestions we have added a couple sentence on role collection can play helping research with biosecurity questions and permitting. See lines 450-453 in the revised manuscript. 

Reviewer 2 Report

The manuscript remarks the role and the importance of culture collections in combating plant diseases by reviewing some case studies that clearly support this hypothesis. The authors, stating and demonstrating a clear decline in collecting, accessioning and funding of culture collections, in particular those relative to  plant pathogenic bacteria, warmly and smartly plan to convince the readers that all isolations and purifications of new as well as old strains should be long-term stored and invite them to deposit in the main culture collections properly listed and suggesting some interesting practices aimed to improve microbial preservation and accessioning rates.

The idea is interesting and focuses the topics of special issue of microorganisms journal, the manuscript is very well written, with a mix of relevant scientific details and a stimulating style, and deserves, in my opinion to be published in the present form.

Author Response

Reviewer 2 has made the day of all authors on this manuscript! We appreciate they time you took to review this work. No changes have been made as none were suggested.